# Can clinical audits be enhanced by pathway simulation and machine learning? An example from the acute stroke pathway

Michael Allen,[1] Kerry Pearn,[1] Thomas Monks,[2] Benjamin D Bray,[3] Richard Everson,[4] Andrew Salmon,[1] Martin James,[5] Ken Stein[1]

¹Medical School, University of Exeter, Exeter, UK
²University of Southampton, Southampton, UK
³Public Health, Royal College Physicians, London, UK
⁴Computer Sciences, University of Exeter, Exeter, UK
⁵Stroke Consultant, Royal Devon & Exeter NHS Trust, Exeter, UK

**Correspondence to**
Dr Michael Allen;
M.Allen@exeter.ac.uk

## ABSTRACT

**Objective** To evaluate the application of clinical pathway simulation in machine learning, using clinical audit data, in order to identify key drivers for improving use and speed of thrombolysis at individual hospitals.

**Design** Computer simulation modelling and machine learning.

**Setting** Seven acute stroke units.

**Participants** Anonymised clinical audit data for 7864 patients.

**Results** Three factors were pivotal in governing thrombolysis use: (1) the proportion of patients with a known stroke onset time (range 44%–73%), (2) pathway speed (for patients arriving within 4 hours of onset: per-hospital median arrival-to-scan ranged from 11 to 56 min; median scan-to-thrombolysis ranged from 21 to 44 min) and (3) predisposition to use thrombolysis (thrombolysis use ranged from 31% to 52% for patients with stroke scanned with 30 min left to administer thrombolysis). A pathway simulation model could predict the potential benefit of improving individual stages of the clinical pathway speed, whereas a machine learning model could predict the benefit of 'exporting' clinical decision making from one hospital to another, while allowing for differences in patient population between hospitals. By applying pathway simulation and machine learning together, we found a realistic ceiling of 15%–25% use of thrombolysis across different hospitals and, in the seven hospitals studied, a realistic opportunity to double the number of patients with no significant disability that may be attributed to thrombolysis.

**Conclusions** National clinical audit may be enhanced by a combination of pathway simulation and machine learning, which best allows for an understanding of key levers for improvement in hyperacute stroke pathways, allowing for differences between local patient populations. These models, based on standard clinical audit data, may be applied at scale while providing results at individual hospital level. The models facilitate understanding of variation and levers for improvement in stroke pathways, and help set realistic targets tailored to local populations.

## INTRODUCTION

NHS England describes clinical audit as a way of identifying whether healthcare is

**Strengths and limitations of this study**

► The pathway simulation and methodology presented enhances the national clinical audit by providing insight into the interhospital variation in use and speed of thrombolysis.
► The methods allow reasonable thrombolysis targets to be set that are tailored to local stroke patient population characteristics.
► The methods (with code published in full) may be applied using open source software and may be applied at scale while providing tailored insight for individual hospitals.
► The methods are limited to information gathered in the national clinical audit for stroke and do not include use of advanced imaging techniques for selection of patients for thrombolysis.

being provided in accordance with agreed standards and where improvements could be made to improve outcomes for patients.[1] Audits may be local or national. In England the Healthcare Quality Improvement Partnership (HQIP), on behalf of the National Health Service (NHS), is responsible for overseeing and commissioning more than 30 clinical audits, which form the National Clinical Audit Programme.[2] These collect and analyse data supplied by local clinicians.

The national audit covering stroke is the Sentinel Stroke National Audit Programme (SSNAP).[3] Stroke is a leading cause of death and disability worldwide, with an estimated 5.9 million deaths and 33 million stroke survivors in 2010.[4] In England, Wales and Northern Ireland, 85 000 people are hospitalised with stroke each year,[5] and stroke is ranked third as a cause of disability-adjusted life years in the UK over the last 25 years.[6]

SSNAP collects longitudinal data on the processes and outcomes of stroke care up to 6 months poststroke for more than 90% of

stroke admissions to acute hospitals in England, Wales and Northern Ireland. Every year data from approximately 85 000 patients are collected. SSNAP publishes quarterly and yearly analysis of results on its website.[3]

SSNAP audit data are used for a wide range of research, such as investigating how the type of stroke affects clinical decisions,[7] how socioeconomic factors influence risk of stroke, care received and outcomes,[8] and how care processes may vary by time of day and day of week.[9]

In this paper we report on the potential of using simulation and machine learning to enhance the output of the SSNAP clinical audits. In particular we focus on the acute stroke pathway and clinical decision making leading to the use of thrombolysis for the treatment of acute stroke, the only licensed drug treatment for acute stroke and one that is critically time-dependent,[10] with little or no benefit after 4.5 hours from stroke onset. The population benefit from thrombolysis has been limited by slow uptake of the treatment and in-hospital delays to the administration of thrombolysis.[11–13]

In England, Wales and Northern Ireland, 11.1% of patients with confirmed acute stroke receive thrombolysis, but use in individual acutely admitting stroke teams varies from 0% to 24.5%.[5] The lowest 10% of acutely admitting stroke teams administer thrombolysis to fewer than 5.9% of patients, whereas the top 10% administer thrombolysis to more than 16.7%. Time from arrival to thrombolysis ('door-to-needle') also varies significantly. The fastest 10% of hospitals have door-to-needle times of 40 min or less, whereas the slowest 10% have door-to-needle times of 85 min or more.[5] There is therefore considerable variation between hospitals in the use and speed of thrombolysis for patients with acute stroke.

The model described here has three components: (1) a clinical pathway model, (2) a clinical outcome model based on the speed and number of patients treated with thrombolysis, and (3) a clinical decision-making model based on machine learning.

Analysis of patient pathway data coupled with pathway modelling has previously allowed investigation and improvement of thrombolysis use in individual hospitals, increasing both the number of patients treated and reducing door-to-needle times.[14 15] These models have usually focused on the speed of the acute stroke pathway from arrival at hospital to giving thrombolysis, and have been tailored to use data available at an individual hospital.[14]

Pathway modelling based on simulating process steps allows for good simulation of the speed of the stroke pathway, but cannot easily model differences in clinical decision making. We were interested in testing whether a model could dissect out the variation in thrombolysis rate that is dependent on differences in patient populations (eg, age or stroke severity) in different hospitals, from the differences that are dependent on the culture of decision making at different hospitals (eg, more cautious vs more aggressive clinical decision making). A variety of machine learning techniques now exist,[16] which are able to make good predictions on pre-existing multidimensional data over a binary or categorical outcome variable (such as whether a patient receives thrombolysis or not). These have the potential to add modelling of clinical decision making to a model of the acute stroke pathway, with the aim of predicting what decision (to thrombolyse or not) would be made for the same patient in different hospitals. Models may also be trained on a reference standard set of hospitals (regarded as centres of clinical excellence) and use of thrombolysis for any patient predicted using that 'benchmark clinical decision-making model'.

We have chosen to combine these three components of pathway, clinical outcome and clinical decision-making modelling as these provide a more powerful and informative model than can be achieved by any single technique.

The aim of our work was to extend previous work on stroke thrombolysis pathway simulation in three significant ways: (1) to create a generic stroke thrombolysis pathway simulation model that could be easily applied to all hospitals in SSNAP; (2) extend the analysis to include factors other than door-to-needle times, with special focus on differences in clinical decision making as analysed and modelled with machine learning techniques; and (3) develop modelling framework that is open source and fast enough to run routine analysis at national level.

## METHODS
### Data
Anonymous SSNAP data were obtained from seven hospitals in England for patients with confirmed stroke over a period of 2 years (2013–2014) for each hospital. These data were secondary data, collected during routine care. No patient identifiable information was obtained.

For the pathway simulation model, the data set contained 7871 patient records with complete data for 12 parameters regarding their characteristics and time-stamped pathway location. These data represent out-of-hospital onset of stroke (which accounts for 94% of all admissions recorded in the SSNAP data used).

For machine learning, only those patients with a completed National Institutes of Health Stroke Scale and who had at least 30 min left to give thrombolysis were used (1862 patients). As a precaution to maintain complete patient anonymity, 17 patients aged under 40 had their age censored to 40, and 6 patients over the age of 100 had their age censored to 100.

### Pathway simulation model
The pathway simulation model (shown schematically in figure 1) simulates the flow of patients through the acute stroke pathway to the point of thrombolysis. Model parameters were set for each hospital by sampling from distributions derived from anonymous data retrieved from SSNAP for each hospital. The model parameters, distribution types and the values used are given in the online supplement.

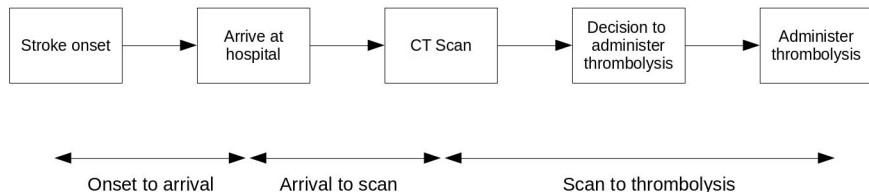

**Figure 1** Modelled sequence of steps of the emergency stroke pathway leading to thrombolysis.

For a patient to receive thrombolysis, they must meet the following criteria: (1) stroke onset time known, (2) arrival at hospital within 4 hours of stroke onset, (3) have an ischaemic stroke and judged to be eligible for thrombolysis, and (4) be within the allowed thrombolysis time window (4.5 hours and 3 hours onset-to-treatment time for patients aged under 80 and 80+, respectively), when summing the process step times in the model. If a patient receives thrombolysis in the model, then the probability of an additional good outcome (Modified Rankin Scale 0–1, no significant disability and able to carry out all usual activities) due to use of thrombolysis is calculated from the onset-to-treatment time and is based on the meta-analysis by Emberson *et al*.[10]

The pathway simulation model was validated by (1) random bootstrap sampling of 100 groups of 600 patients with varying overall thrombolysis use, comparing actual with predicted thrombolysis use, and (2) comparing actual and predicted thrombolysis use and speed across the seven hospitals.

### Clinical decision model (machine learning)

The clinical decision model aims to replicate the decision to give or not give treatment with thrombolysis for any given patient at any given hospital.

The model predicts whether an individual patient would receive thrombolysis or not from a set of 50 parameters defining the patient's characteristics, clinical well-being and hospital attended. Following a comparison of methods, a random forests method was chosen (see online supplement for a list of all features used and a comparison of different machine learning models).

This model is intended to make decisions based only on clinical presentation, assuming that there is time to give thrombolysis. Patients were included if they had been scanned with 30 min left to give thrombolysis (allowing 4.5 hours and 3 hours from onset to treatment for patients aged under 80 and 80+, respectively).

The machine learning model was validated using stratified tenfold validation, where the data are split into 10 subsets, and the model run 10 times (with each model run using 9 subsets for training and 1 subset held back for testing, with all data present in a test subset once and only once). All machine learning methods and validation were coded in Python using the SciKit Learn machine learning library.[17]

### Patient and public involvement

Through this study we have used a panel of four to five stroke survivors or carers of stroke survivors. These were recruited through the National Institute for Health Research PenCLAHRC (The National Institute for Health Research Collaboration in Applied Health Research and Care South West Peninsula) Patient and Public Involvement group. These have met three times during the course of this project to help review aims, results and future plans.

### RESULTS
### Pathway simulation

The pathway simulation model was validated by (1) random bootstrap sampling varying overall thrombolysis use, comparing actual with predicted thrombolysis use,

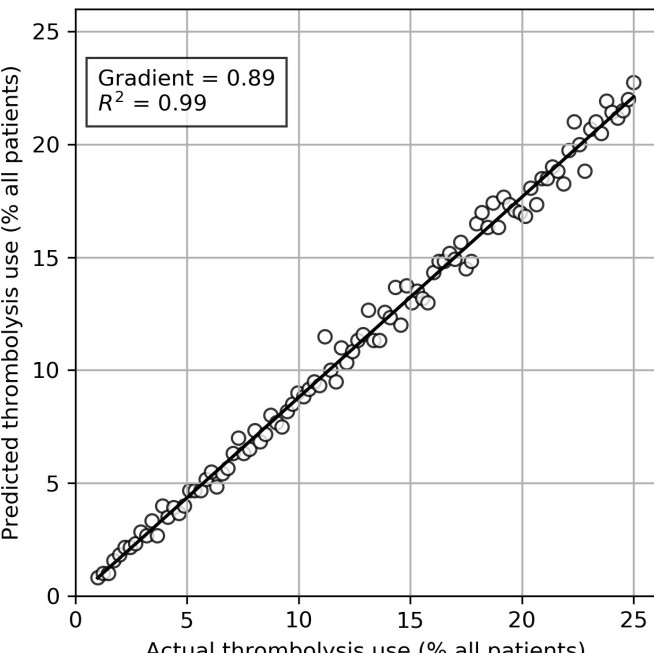

**Figure 2** Validation of the pathway simulation model, comparing actual with modelled (predicted) thrombolysis based on random sampling of all data. Samples were 600 points (representing typical acute stroke unit admission numbers) chosen randomly with resampling from patients given or not given thrombolysis to create a range of thrombolysis use examples. Points show mean predicted thrombolysis use from 100 runs, with each run modelling 1 year.

**Table 1** Comparison of actual versus modelled hospital performance

| Hospital | 1 | 2 | 3 | 4 | 5 | 6 | 7 |
|---|---|---|---|---|---|---|---|
| Actual thrombolysis (%) | 13.7 | 8.0 | 8.4 | 7.0 | 8.5 | 14.5 | 9.2 |
| Model thrombolysis (%) | 12.9 | 7.1 | 7.6 | 6.3 | 8.1 | 12.0 | 8.0 |
| Actual onset to thrombolysis (hours) | 2.4 | 2.6 | 2.9 | 2.5 | 2.3 | 2.5 | 2.5 |
| Model onset to thrombolysis (hours) | 2.6 | 2.6 | 3.0 | 2.6 | 2.7 | 2.6 | 2.5 |

and (2) comparing actual and predicted thrombolysis use and speed across the seven hospitals.

In the random sampling (figure 2), 600 patients (typical of an acute stroke unit) were sampled randomly, with resampling from either the thrombolysis group or the no-thrombolysis group to give varying samples with an overall thrombolysis use rate of 1%–25%. There were 100 model runs performed. The model showed very good correlation ($R^2$=0.99) between actual and predicted values, although the model slightly underpredicted actual thrombolysis use, with predicted thrombolysis being, on average, 89% that of actual thrombolysis use.

The model was further validated by comparing modelled (predicted) use of thrombolysis with actual use of thrombolysis (table 1). Actual use of thrombolysis was based on modelling a 1-year period with replicates of 100 runs (each with different random number seeds) in order to determine expected year-to-year variation. The model showed very good correlation ($R^2$=0.96) between actual and predicted values, although the model again slightly underpredicted actual thrombolysis use, with predicted thrombolysis being, on average, 90% that of actual thrombolysis use.

The difference between predicted and actual thrombolysis use is largely explained by the observation that 8% of thrombolysis in the SSNAP data set was given outside of the assumed allowable times for thrombolysis in the model; the model applies a stricter time cut-off than clinicians allow in reality.

The model was run with various 'what-if?' scenarios for each of the seven hospitals (table 2).
► Base case: model based on parameters derived frm current hospital-specific performance
► Scenario B: 60% of patients with ischaemic stroke scanned with 30 min left to treat receive thrombolysis (an analysis of ECASS-3/IST-3 results concluded that 591 out of 992, or 60%, of patients with ischaemic stroke arriving within 4 hours of stroke onset were suitable for thrombolysis[18]).
► Scenario C: onset time known fixed at 77% (national SSNAP upper quartile for year 2015/2016[5]).
► Combination of above.

In order to achieve the greatest improvement in thrombolysis use in each of the seven hospitals, for two hospitals (hospitals 1 and 6) it would be best to improve the speed of the pathway, for two hospitals (hospitals 4 and

**Table 2** Predicted thrombolysis use and clinical benefit across all modelled hospitals (1–7)

| | Hospital | Base | A | B | C | ABC |
|---|---|---|---|---|---|---|
| Thrombolysis use (%) | 1 | 12.7 (0.3) | 17.6 (0.3) | 13.9 (0.3) | 14.6 (0.3) | 21.5 (0.3) |
| | 2 | 7.0 (0.2) | 10.4 (0.3) | 10.8 (0.2) | 10.0 (0.3) | 23.4 (0.3) |
| | 3 | 7.7 (0.2) | 9.7 (0.2) | 11.9 (0.2) | 10.9 (0.2) | 21.7 (0.3) |
| | 4 | 6.2 (0.2) | 6.8 (0.2) | 11.2 (0.2) | 11.3 (0.3) | 20.7 (0.3) |
| | 5 | 7.9 (0.3) | 10.4 (0.3) | 10.0 (0.3) | 13.2 (0.4) | 22.1 (0.5) |
| | 6 | 12.3 (0.3) | 16.3 (0.4) | 14.5 (0.4) | 12.7 (0.3) | 20.2 (0.4) |
| | 7 | 8.0 (0.2) | 9.4 (0.2) | 14.0 (0.3) | 10.8 (0.2) | 22.5 (0.3) |
| Additional good outcomes per 1000 admissions | 1 | 11.1 (0.2) | 17.2 (0.3) | 12.2 (0.2) | 12.8 (0.3) | 21.0 (0.3) |
| | 2 | 6.1 (0.2) | 10.2 (0.3) | 9.3 (0.2) | 8.6 (0.2) | 23.0 (0.4) |
| | 3 | 6.5 (0.2) | 10.1 (0.2) | 10.0 (0.2) | 9.1 (0.2) | 22.5 (0.3) |
| | 4 | 5.4 (0.2) | 6.7 (0.2) | 9.8 (0.2) | 9.9 (0.2) | 20.0 (0.3) |
| | 5 | 7.0 (0.3) | 10.8 (0.4) | 8.7 (0.3) | 11.8 (0.4) | 23.1 (0.5) |
| | 6 | 10.6 (0.3) | 16.0 (0.4) | 12.5 (0.3) | 11.0 (0.3) | 19.9 (0.4) |
| | 7 | 7.1 (0.2) | 9.5 (0.3) | 12.4 (0.2) | 9.6 (0.2) | 22.9 (0.3) |

Data show (base) model based on parameters derived from current performance; (A) arrival-to-scan and scan-to-thrombolysis both fixed at 15 min (with no variation in either time); (B) judged to be eligible for thrombolysis fixed at 60%; (C) onset time known fixed at 77%; and combinations of the above. Results show mean and ±95% confidence limits (100 runs).

**Table 3** Actual and predicted thrombolysis use (for patients scanned with time left to thrombolysis) if the decision to give thrombolysis is based on decisions made by a random forest model trained at different hospitals

| Actual thrombolysis use by hospital | | | | | | | |
|---|---|---|---|---|---|---|---|
| | 1 | 2 | 3 | 4 | 5 | 6 | 7 |
| | 52 | 35 | 48 | 33 | 49 | 44 | 31 |

Predicted thrombolysis use at each hospital depending on which hospital is used to train the decision model and which hospital patients actually attend

| | | Hospitals patients actually attend | | | | | | |
|---|---|---|---|---|---|---|---|---|
| | | 1 | 2 | 3 | 4 | 5 | 6 | 7 |
| Hospital used to train model | 1 | 52 | 42 | 58 | 50 | 67 | 57 | 45 |
| | 2 | 48 | 35 | 55 | 36 | 46 | 37 | 29 |
| | 3 | 53 | 38 | 48 | 46 | 58 | 41 | 34 |
| | 4 | 40 | 28 | 48 | 33 | 52 | 29 | 26 |
| | 5 | 50 | 36 | 50 | 40 | 49 | 45 | 37 |
| | 6 | 49 | 32 | 55 | 44 | 59 | 44 | 39 |
| | 7 | 42 | 23 | 42 | 31 | 50 | 36 | 31 |

The columns represent the likely difference in thrombolysis use due to differences in decision making.

5) it would be best to improve determination of stroke onset time, and for three (hospitals 2, 3 and 7) it would be best to judge more patients as eligible for thrombolysis for those scanned with time left to treat. If a priority is to maximise clinical outcome, then for four hospitals (hospitals 1, 2, 3 and 6) it would be best to improve the speed of the pathway, for two hospitals (hospitals 4 and 5) it would be best to improve the determination of stroke onset time, and for one (hospital 7) it would be best to judge more patients as eligible for thrombolysis for those scanned with time left to treat. Combining all changes in the model could produce thrombolysis rates up to 20%–23%, and 20–23 additional non-disabled outcomes per 1000 patients admitted with stroke.

In the case of hospital 5, arrival-to-scan times could be slowed by an average of 30 min and clinical outcomes would still be greater if that hospital achieved a proportion of known stroke onset time equal to the national average.

### Clinical decision (machine learning) model

The random forest mode chosen has an 82% accuracy in predicting whether a patient received thrombolysis or not (see online supplement for more details on validation). A machine learning model may be trained on a subset of patients to investigate how the difference in thrombolysis use between hospitals may be proportionally attributed to either the hospital or the local patient population. Table 3 shows the predicted use of thrombolysis in a set of patients that attend one hospital, based on decisions made from training at another hospital. Taking hospital 7 as an example, between 26% and 45% of the patients

who currently attend hospital 7 (with time left to receive thrombolysis) might receive thrombolysis depending on which hospital decision making is used to train the model. Patient cohort also affects the predicted thrombolysis use. Taking hospital 7 as an example again, if the model is trained on decisions made for patients attending hospital 7, and different hospital patient groups are then analysed in the model, then thrombolysis use is predicted to be between 23% and 50% depending on the admitting hospital patient group analysed.

### Combining pathway simulation and machine learning

The output from machine learning may be incorporated into the stroke pathway model by using the machine learning model to make the decision in the pathway model about whether a patient is 'judged to be eligible for thrombolysis (for patients scanned with 30 min left to administer thrombolysis)'. This should tailor the clinical decision to the local population, without being affected by any particular hospital's predisposition to use thrombolysis. The 'judged to be eligible for thrombolysis' parameter in the pathway model may take its value from a machine learning model trained using a reference set of hospitals. The clinical decision making from these reference hospitals may be used to predict which of the patients from the hospital under study are eligible for thrombolysis.

Table 4 compares base case hospital performance (predicted thrombolysis use and clinical benefit) with the performance obtainable by a new realistic 'alternative' practice which is in part informed by the random forest machine learning model: (1) the proportion of patients with a known stroke onset time is set at the national median (67%) unless a hospital is already higher; (2) the door-to-needle time is set to 40 min for 90% of patients (20 min arrival-to-scan and 20 min scan-to-needle), with the other 10% of patients not receiving a scan within 4 hours of arrival; and (3) the clinical decision to administer thrombolysis for those patients scanned with 30 min left to treat is set by the machine learning model trained from a reference hospital (this example uses the hospital that has the highest use of thrombolysis for those patients scanned with time to treat). Resulting thrombolysis targets vary from 16% to 25% depending on the hospital (base case 6%–13%).

### DISCUSSION

While there is no agreed benchmark for the proportion of patients with stroke who should receive thrombolysis, it has been suggested that about half of patients with stroke, if they arrive at hospital in time, could be clinically suitable for thrombolysis.[18] In practice the greatest proportion of patients treated with thrombolysis in English hyperacute stroke services is close to 20%.[5] These figures contrast with an average 12% across England and Wales[5] and a range of 6%–14% across the seven acute stroke centres in our study. Internationally thrombolysis rates also appear

**Table 4** Combining pathway simulation and machine learning

| Hospital | Thrombolysis use (%) | | Additional good outcomes per 1000 admissions | |
|---|---|---|---|---|
| | Current | Alternative | Current | Alternative |
| 1 | 12.9 (0.3) | 18.6 (0.3) | 11.3 (0.2) | 17.4 (0.3) |
| 2 | 7.1 (0.2) | 15.3 (0.3) | 6.1 (0.2) | 14.6 (0.3) |
| 3 | 7.6 (0.2) | 22.0 (0.3) | 6.3 (0.1) | 21.4 (0.3) |
| 4 | 6.3 (0.2) | 17.2 (0.3) | 5.5 (0.2) | 16.0 (0.3) |
| 5 | 8.1 (0.3) | 25.3 (0.5) | 7.1 (0.3) | 25.2 (0.5) |
| 6 | 12.0 (0.4) | 23.4 (0.4) | 10.5 (0.3) | 22.0 (0.4) |
| 7 | 8.0 (0.2) | 16.9 (0.3) | 7.1 (0.2) | 16.5 (0.3) |

Predicted thrombolysis use and clinical benefit (additional good outcomes per 1000 admitted patients) across all modelled hospitals (1–7) from the pathway simulation. Data show (base) model based on parameters derived from current performance; alternative 'realistic target' settings, fixing the proportion of known stroke onset times to the national SSNAP average (67% median) unless the hospital currently performs higher, fixing arrival-to-scan and scan-to-needle to 20 min each (with 10% of patients not scanned within 4 hours), and fixing the proportion of treatable patients (scanned with 30 min left to treat) according to the output of the machine learning model based on the hospital with the maximum predicted proportion given thrombolysis. Data show mean and 95% CI.
SSNAP, Sentinel Stroke National Audit Programme.

on average to be lower than best-practice centres: for example, rates have been reported to be 14.6% in the Netherlands in 2012[19] and 5.4% in USA in 2010[20] (for patients aged 65 years or more).

Our pathway simulation model slightly underpredicted actual use of thrombolysis. This was mostly due to the model applying a strict cut-off of allowable time from onset of stroke to giving thrombolysis. In a real clinical setting a little bit of flexibility may be applied.

As patients move through the stroke pathway, an unknown stroke onset time would be the first key barrier to thrombolysis in our model—a barrier to thrombolysis previously noted.[21] Although a qualitative analysis of methods for determining the onset time of a stroke was beyond the scope of this study, it was clear that hospitals differed in how this information was gathered, such as whether they relied on information from paramedics only or whether the hospital clinician would also investigate an unknown stroke onset time. In our modelling study we looked at the potential impact of reaching the national upper quartile for ascertaining stroke onset times. In one of the seven hospitals, this factor was the single largest in attaining improvements to the thrombolysis rate. For patients with unknown time of onset, it may be possible to use advanced scanning methods to estimate stroke onset times and suitability for treatment[21] and further increase the population eligible for thrombolysis treatment. As advanced imaging techniques become established, the model could/should be extended to include this alternative pathway which our model does not currently include.

Reductions to in-hospital treatment delays have been the focus of previous modelling approaches,[14 15] and the 'need for speed' has frequently been stressed.[11 12] Our study found that compared with current state, it was reasonable to expect that improvements to pathway speed and reliability could achieve close to 50% increase in the number of disability-free patients. Our modelling was

based on consistently achieving 30 min door-to-needle times. It may be possible to be even more aggressive on process time as speeds of 20 min door-to-needle times have been reported.[22] Those hospitals in our study where paramedics took FAST-positive patients straight to the scanner (bypassing emergency department) had significantly faster arrival-to-scan times, and this may be a more widely applicable approach, although it was not responsible on its own for a higher thrombolysis rate in those centres, illustrating the multifactorial influences on both thrombolysis rate and door-to-needle time; it may sometimes be best to accept slower pathway speeds in order to improve other factors in the pathway (such as ascertaining stroke onset time). A challenge for smaller hospitals is to have consistently fast processes 24 hours per day, 7 days per week; scanners must be rapidly available for patients with stroke at any time of day and day of week.

The final hurdle investigated was in the confidence in using thrombolysis for patients scanned in time. Use of thrombolysis, when time allowed, varied significantly between hospitals. Physician uncertainty or lack of confidence in thrombolysis has previously been identified as a barrier to use of thrombolysis.[23] Despite meta-analysis demonstrating the clinical benefit of thrombolysis,[10] the use and benefit of thrombolysis have still recently been under scrutiny.[24] This could certainly explain the large between-hospital variation in thrombolysis rates we observed for those patients scanned with time left to receive thrombolysis.

Scenarios modelled using our stand-alone pathway model used a standard 'optimum' rate for the clinical decision to use thrombolysis based on an analysis of the IST-3 trial. However this scenario assumes all hospitals receive patients who have similar characteristics overall to those in the IST-3 trial, and assumes that the IST-3 trial criteria should be a benchmark for all hospitals. The machine learning model allows for differences in patient

populations between hospitals (eg, reflecting different age demographics between different regions) and allows for differences in real-life (rather than clinical trial) decision making. Using results from the machine learning model, which focuses solely on clinical suitability for thrombolysis when there is time left to treat after the scan, enabled the pathway model to be refined and tailored further to local hospital populations. Machine learning models had an accuracy of 80%–82% in predicting decisions made. The current performance, while falling short of performance necessary for real-world decision making or guidance, is still useful in audit to understand variation in use of thrombolysis.

Using machine learning algorithms we found that differences in thrombolysis rate, for patients with time left to thrombolyse after the scan, can be explained by differences in patient population and in differences in decision making between hospitals. Rather than being based on a set of simple rules, our machine learning model learns from the clinical decision in real-life settings. A machine learning model allows prediction of potential thrombolysis use given the characteristics of the local population, but based on real-life clinical decision making at a range of hospitals, identifying those hospitals which appear to be outliers in their decision making. Bembenek *et al*[18] have estimated that if the inclusion criteria of the IST-3 trial are used to determine patients suitable for thrombolysis, about 60% of patients with ischaemic stroke (equivalent to 50% all patients with stroke, if 85% of patients have ischaemic stroke) could be suitable for thrombolytic treatment if they arrive in time and the pathway is efficient. This overall figure however does not allow for differences in patient mix in different geographical areas. A clinician reviewing their own use of thrombolysis may conclude that their use is different because their patients are different. Machine learning confirms differences in patient populations, but also suggests differences in clinical decision making. By applying machine learning, feedback may be given regarding their use of thrombolysis compared with other hospitals, allowing for differences in local populations. This may be especially useful if a benchmark group of hospitals (acknowledged centres of excellence in stroke care) is used to train the model.

When applying 'stretch targets' (eg, basing overall use of thrombolysis on an analysis of the IST-3 trial) to stroke pathways in the pathway model, we found that maximum thrombolysis use could be about 20% to 23% of all emergency-confirmed stroke admissions. This could have the benefit of producing another 20-23 people with no significant disability for every 1000 patients admitted with confirmed stroke, with potential to also improve outcomes for those who cannot be classified as having no significant disability even after use of thrombolysis. Using slightly less challenging targets for the acute stroke pathway process and using machine learning to mimic decisions to thrombolyse adjusted for local populations, we found that individual hospital targets could realistically be set in the 16%–25% range, which would lead to

another 16–25 people with no significant disability for every 1000 patients admitted. The upper limit varied between hospitals based on the patient mix attending that hospital. Such results warn against dangers of setting any universal expectation of use of thrombolysis.

This approach has the potential to be applied locally or, as it is based solely on SSNAP data, as part of the national audit of stroke services. The methods have the potential to be automated, allowing for incorporation into the quarterly stroke service audit conducted by SSNAP. Where clinical decision making appears to be significantly different from a reference group of hospital, the machine learning model may be used to identify a small group of patients for review, where clinical decision making appears to be different from a model trained on an agreed benchmark set of hospitals.

Our work is not the first work to apply simulation or machine learning in stroke. Previous work has been published on using pathway simulation at an individual hospital level.[14 15] We build on previous work by using standardised data and an open simulation framework to enable pathway simulation to be performed at scale across all hospitals covered by the national stroke audit. Previous work has also been published using machine learning to predict risk of stroke[25] and likely outcome.[26–28] Where our work adds novelty is first in the combination with pathway simulation and the use of machine learning to investigate differences in clinical decision making between hospitals, and to apply 'what if?' scenario testing of what might the likely clinical benefit be of standardising clinical decision making in accordance with recognised centres of excellence, taking into account differences in local patient populations.

### Strengths and weaknesses

A key strength of our study is that, since the models are based solely on SSNAP data, the models developed may be applied to all UK hospitals. The use of open source software and the provision of our code in a public repository should also facilitate easier adoption. Monks *et al*[29] have noted that when modelling stroke pathways, the level of detail of the model should depend on the questions being addressed. In our approach we consciously limit ourselves to the detail found in SSNAP data. A weakness, therefore, is that the models as they are presented cannot address questions that require data outside of the SSNAP data set. Our models highlight the improvements that can be achieved by an overall pathway stage (such as arrival-to-scan) but do not inform how improvement in that step could/should be achieved.

A strength of our modelling compared with normal approaches to stroke pathway modelling[29] is that we have used machine learning in order to both allow for differences in local stroke patient populations (eg, differences in age, gender and severity) and to understand differences in decision making between hospitals. A limitation of this approach is that the modelling is limited to hospital level and will not uncover differences in use of thrombolysis

by different practitioners within the same hospital. Nevertheless, the model will pick up overall organisational cultural attitudes towards use of thrombolysis in stroke (such as whether one hospital is more or less likely to use thrombolysis for patients with less severe stroke).

The use of open source software and the provision of our code in a public repository should also facilitate easier adoption or refinement of the approach by others.

A weakness of this study is that it was limited in scope to seven hospitals. This methodology should preferably be validated on a larger data set (eg, the national SSNAP data set).

Our model is deliberately limited to the SSNAP data set, as that is universally available across all stroke units in England and Wales. There are likely useful clinical features missing from the SSNAP data set (eg, information from advanced imaging). As the SSNAP data set grows, we would expect accuracy of the machine learning model to increase beyond the current 80%, but we believe the model is already sufficiently useful to add more insight into the routine SSNAP audit.

In this model we have considered only use of thrombolysis. As the SSNAP data set grows, we would expect a similar approach to be useful in identifying levels to increase use and speed of thrombectomy (by introducing interhospital transfer, when required, into the clinical pathway model, and also building a learning model on use of thrombectomy to identify differences in selection of patients for thrombectomy). Additionally, we have focused on the most usual path to thrombolysis—those arriving and scanned with time left to treat within the normal thrombolysis time limits. Advanced imaging may be used to select further patients[30]; as advanced imaging becomes more commonplace, then the model would be best extended to include this additional pathway.

In this study we have not focused on how implementation of changes may be brought about in different hospitals. It is possible that some improvements in the pathway (such as rapid availability to scan 24/7) may only be possible by centralisation of services into larger units who may have more resources to deploy.[31–33] Our model will show the benefit of reducing pathway speed and variability, but does not imply such a change is necessary and always possible in smaller hospitals.

## CONCLUSIONS

The stroke clinical audit reports on thrombolysis usage (percentage of patients receiving thrombolysis) and time to thrombolysis. Interhospital variation in use of thrombolysis may be due to (1) differences in stroke pathway, (2) differences in patient population characteristics or (3) differences in clinical decision making. In our study we present a method that qualitatively analyses key components of the stroke pathway in a manner that may be applied at scale while producing individualised results for each hospital, highlighting how thrombolysis use and speed may best be improved. We believe this type of approach will augment current outputs from national clinical audits.

**Contributors** MA, KP, MJ and KS conceived the study. RE reviewed and provided guidance on the machine learning aspects. TM and KP reviewed and provided guidance on the simulation aspects. AS conducted literature reviews and provided guidance on methodologies. MA wrote the algorithm code used in this study and performed the primary analysis. MJ and BDB informed the base assumptions of the model, provided key literature and reviewed the output. MA produced the first draft of the paper, and all authors contributed significant editing to the paper. KS oversaw all stages of the project. KS is the guarantor.

**Funding** This study was jointly funded by the National Institute for Health Research (NIHR) Collaboration for Leadership in Applied Health Research and Care for the South West Peninsula, and the South West Academic Health Science Network.

**Disclaimer** The views and opinions expressed in this paper are those of the authors, and not necessarily those of the NHS, the National Institute for Health Research or the Department of Health.

**Competing interests** None declared.

**Patient consent for publication** Not required.

**Ethics approval** This study used anonymous secondary patient data only, collected as part of routine care audit. The study was conducted as part of a local service evaluation and improvement exercise sponsored by the South West Strategic Clinical Network. There was no access to patient identifiable information, no collecting of primary data, and no patient care was altered during the process of this study. There was no patient enrolment into the study. A decision that no formal ethical approval was needed was made by the project steering group with representation from the South West Strategic Clinical Network, the National Institute for Health Research (NIHR) Collaboration for Leadership in Applied Health Research and Care for the South West Peninsula, and the South West Academic Health Science Network.

**Provenance and peer review** Not commissioned; externally peer reviewed.

**Data availability statement** Data are available in a public, open access repository.

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
