## [Reviewer comments · BMJ Open]

ARTICLE DETAILS

TITLE (PROVISIONAL)	Can clinical audits be enhanced by pathway simulation and machine learning? An example from the acute stroke pathway.
AUTHORS	Allen, Michael; Pearn, Kerry; Monks, T; Bray, Benjamin D.; Everson, Richard; Salmon, Andrew; James, Martin; Stein, Ken

VERSION 1 – REVIEW

REVIEWER	Paul Bentley Imperial College London
REVIEW RETURNED	16-Jan-2019

GENERAL COMMENTS	The study represents a meaningful and potentially relevant approach to analysing multidimensional audit data. A pathway simulation approach is shown to be particularly suited to outcomes where these depend upon effects at multiple interconnected steps. This could be used to improve service provision. However several parts are deficient. There should be a more thorough rationale provided at the outset as to why the chosen methods are most suitable to answer the relevant audit questions. Why is multiple regression inadequate? Given this journal has broad readership, it should be explained more clearly to a non-statistician how a pathway simulation model is set up and run. Details could be provided in the appendix. This is especially so since the authors kindly offer to share their software. In the pathway simulation validation, the model parameters should be derived from data independent (eg different years) to that from which it is being tested upon. Can the authors confirm this or explain why not? Can the authors also validate overall Onset -to-Thrombolysis time ? This would avoid the confound suggested for Thrombolysis Decision Rate that doctors may select extra cases, who are outside standard inclusion criteria. Are parameters in the simulation model independent? If not, how does one know which are the instrumental ones that should be targeted by intervention, and which merely mirror effects elsewhere? Can the machine learning model be validated? This may require splitting data within rather than between hospitals.
--

	Are there differences in the 50 chosen parameters between hospitals? Could this explain the differences in use. E.g differences in severity or time from onset may differ between hospitals. Specific points in text: 'The pathway simulation model is coded in Python/NumPy and is based on sampling from distributions based on real-world data': The second aspect of the sentence seems vague, What data they are referring to? if SSNAP then I think it needs to be re worded, if not there is no reference. 'For machine learning, only those patients with a completed NIH Stroke Scale and who had at least 30 minutes left to give thrombolysis were used (1,862)': As far as I can tell even in the appendices there is no actual breakdown of the data. So no tabulated ranges for the machine learning model such as age 17-90 etc. So it is hard to say whether this smaller dataset used for this section is actually representative. They trained the model with this data based on the exclusion criteria above, but we don't know whether the dataset they are left with is actually a good representative. 'the machine learning model allows for differences in patient populations between hospitals (for example reflecting different age demographics between different regions), and allows for differences in real life (rather than clinical trial) decision making' : Without having any idea of the original data variation it is hard to justify this point.
--	--

REVIEWER	Haris Kamal University of Texas at Houston, USA
REVIEW RETURNED	31-Jan-2019

GENERAL COMMENTS	The authors present a method to qualitatively analyze the application of clinical pathway simulation in machine learning using real-time data to identify important factors to improve use and timing of thrombolysis in acute stroke care hospitals. The models tried appear to be a good start but perhaps the authors could further validate these models with external data sources- a point which should be mentioned in the discussions section.
---

REVIEWER	Andre Kemmling Department of Neuroradiology, University Hospital Munster, Germany
REVIEW RETURNED	01-Feb-2019

GENERAL COMMENTS	Authors use pathway simulation in acute stroke triage with regard to completion of iv thrombolysis. 7864 records with 12 parameters were analyzed. Authors also use machine learning for iv thrombolysis prediction (1862 records with 50 parameters). Included parameter primarily describe onset time intervals, severity of symptoms and comorbidities. Results show, how onset time intervals affect thrombolysis use in prediction model. across 7 Hospitals. Results and conclusion may be regarded as proof of concept. Major variables are missing to actually relate to stroke triage reality and the list of parameters is far too simplistic to draw conclusions.
---

	Any variable that encodes potential iv rTPA contraindication would affect the model. Imaging parameters such as extent of the early infarct (e.g. by ASPECTS) may significantly affect thrombolysis rates. Also presence of large vessel occlusion may be relevant. Authors should address these limitations extensively and explain how these may be overcome and then improve their method. Authors should use more appropriate up to date references, e.g. 22 Aoki J, Kimura K, Iguchi Y, et al. FLAIR can estimate the onset time in acute ischemic stroke patients. J Neurol Sci 2010;293:39–44. doi:10.1016/j.jns.2010.03.011 references by Thomalla et al are more extensive and recent, in particular with respect to recent NEJM trials with use of MRI to initiate thrombolysis with unknown time of onset.
--	---

REVIEWER	Vikram Puri Researcher, DuyTan University, Vietnam
REVIEW RETURNED	06-Mar-2019

GENERAL COMMENTS	1. Overall Paper is well written but it require two following revisions: a. need more references and paper because authors merged the Introduction and related studies on one section b. They have applied the machine learning and pathway learning so they have to append the results and also some mathematical parameters to increase the visibility of results
--

REVIEWER	Ana Timóteo Santa Marta Hospital, Lisbon, Portugal
REVIEW RETURNED	23-Apr-2019

GENERAL COMMENTS	I have been asked to review specifically the statistics section. In this paper, there is no specific statistical section. Statistics are very briefly explained in the data and results section. The methodology used by the authors is not usual and it requires a more detailed explanation for the common reader to understand it. It is not clear at all how do the authors get to the figures presented in the paper. A recommend a detailed statistics section to be included.
---

REVIEWER	George Vassilacopoulos UNIVERSITY OF PIRAEUS, GREECE
REVIEW RETURNED	04-May-2019

GENERAL COMMENTS	This paper reports on the potential of using simulation and machine learning to enhance the output of the clinical audits recorded by the Sentinel Stroke National Audit Programme (SSNAP) of the NHS England. In particular, the paper focuses on the acute stroke pathway and clinical decision making leading to the use of thrombolysis for the treatment of acute stroke, the only licensed drug treatment for acute stroke, and one that is critically time-dependent with little or no benefit after 4.5 hours from stroke onset. Essentially, the paper extends previous work on stroke thrombolysis by using both pathway simulation and machine learning modelling to include factors other than door-to-needle
---

	times, with special focus on differences in clinical decision making as analysed and modelled with machine learning techniques, so that to develop an open source modelling framework that is fast enough to run routine analysis at national level. The paper is very well presented, with regard to the sound methods (both simulation and machine learning modelling) used and to the thorough discussion cited, for those interested in the particular clinical field. In addition, it is a significant contribution to the literature on the wider field of healthcare analytics using national registry and/or EMR data. Therefore, I would strongly recommend that the paper is accepted for publication in its present form.
--	---

VERSION 1 – AUTHOR RESPONSE

Reviewer(s)' Comments to Author:

Reviewer: 1

Reviewer Name: Paul Bentley

Institution and Country: Imperial College London

Please state any competing interests or state 'None declared': None declared

Please leave your comments for the authors below

The study represents a meaningful and potentially relevant approach to analysing multidimensional audit data. A pathway simulation approach is shown to be particularly suited to outcomes where these depend upon effects at multiple interconnected steps. This could be used to improve service provision.

However several parts are deficient.

There should be a more thorough rationale provided at the outset as to why the chosen methods are most suitable to answer the relevant audit questions. Why is multiple regression inadequate?

More detail added to the introduction. Essentially no single technique can provide a full model, which is why we have taken a more modular approach based initially on a pathway model (which has a published track record of use and benefit in this area). Pathway models have an advantage that, because they model individual steps in a patient journey, clinicians can recognise the basic building blocks of the model. A 'simple' multiple regression model would have the drawback that it will struggle with the non-linear relationships and distributions in our pathway (such as the timebounded cut-off to treatment), and will likely be unreliable for exploring circumstances outside of the range of the regression input data. Pathway simulation models also allow for testing of 'what if?' scenarios that are outside of the range of the data of current performance. Though extrapolation always carries a risk, simulation has been developed to predict performance of changed or even new systems where there is no current data to train a regression model. The machine learning model has been added to the pathway simulation model in order to provide a clinical decision model that allows for differences in both patients and hospital culture in thrombolysis decisions.

Given this journal has broad readership, it should be explained more clearly to a non-statistician how a pathway simulation model is set up and run. Details could be provided in the appendix. This is especially so since the authors kindly offer to share their software.

Thank you. We have added to the methods and have provided detail on the appendix.

In the pathway simulation validation, the model parameters should be derived from data independent (eg different years) to that from which it is being tested upon. Can the authors confirm this or explain why not?

We agree with the reviewer that validation of the model is important. There are some subtleties in dynamic computer simulation that distinguish it from a classic "train/test/validation" split that the reviewer describes above. The scenario described by the reviewer is what we would use if the purpose of the study was to forecast a future event – for example, if we had built a statistical machine learning model to predict n time periods ahead we would use an appropriate rolling forecast window approach for cross-validation. In the computer simulation study we present it is appropriate to think of

the purpose of the model as an evaluation of 'what-if we did things differently?' (or more formally competing system designs) relative to a baseline. We are modelling what would happen if one set of the models inputs (e.g. the clinical team) has changed. This is a traditional approach to computer simulation and is consistent with procedures presented in wellknown gold-standard texts on discrete-event simulation (DES; most notably Robinson, 2014; Law, 2014 [1,2]). These 'what-ifs' modelled in the DES are different worlds that don't exist in the real world and there is no a-priori cross-validation that we can perform. The appropriate approach is to focus on what is called 'white box' validation where the logic, and sun-models, of the simulation is scrutinised and tested. This in part includes the validation of the ML models that we include in the online appendix (though the machine learning models also include a classical train/test split). Another subtle point is that if the inputs to the model changed – for example the patient arrival process then we would need to re-validate the model outputs. So in a production version of our study the issue is not – “we should use different input data for validation” - but instead identifying if there are instances where input data has changed and then automatically revalidating the simulation model.

However we have sought to extend the validation, especially because it was limited to seven data points (the seven hospitals). We have now added validation based on random bootstrap sampling with samples chosen to have different thrombolysis rates in the range in which we are interested. The paper now includes the graph based on sampling, and the hospital-specific results have been transferred to a table which now includes additional comparisons between model and real world (see next section).

Can the authors also validate overall Onset -to-Thrombolysis time ? This would avoid the confound suggested for Thrombolysis Decision Rate that doctors may select extra cases, who are outside standard inclusion criteria.

Thank you for the suggestion. This has been added to the results. Times are very similar. Out of this also came a useful analysis of what proportion of patients receive thrombolysis inside of guideline times. 92% of patients receive thrombolysis within guideline times. The model applies those time guidelines strictly, and this difference explains nearly all of the 11% under-estimation of thrombolysis use in the pathway model (this explanation was put as a possible cause in the original data, but is now backed with analysis).

Are parameters in the simulation model independent? If not, how does one know which are the instrumental ones that should be targeted by intervention, and which merely mirror effects elsewhere? Thank you for the suggestion. We have added a correlation matrix to the appendix. Of the key steps in the model (arrival to scan time, scan to thrombolysis, proportion of patients receiving thrombolysis if there is time to treat), the strongest correlation (R-square) is -0.16.

The model breaks down the pathway into a series of process steps and a decision model, each of which may be changed to improve performance. The main principle of the model is to identify the step(s) which makes most difference to the outcome, thereby identifying the best place to focus attention.

Can the machine learning model be validated? This may require splitting data within rather than between hospitals.

We have added more detail to the validation method in the supplement (it had somehow got lost, sorry!). The machine learning models were validated using stratified ten-fold validation, each of which would split the data within hospitals (the data is split into 10 training/test sets with each patient being in the test set once and only results. The appendix contains the accuracy measures for each method (accuracy, sensitivity, specificity, ROC), along with the ROC plot and learning rate for the chosen method (Random Forest).

Are there differences in the 50 chosen parameters between hospitals? Could this explain the differences in use. E.g differences in severity or time from onset may differ between hospitals.

Yes, and that is indeed the purpose of the machine-learning model – to provide a 'target' thrombolysis use considering the mix of patients in any given hospital. So, for example, if one hospital has more severe or older patients then the machine learning model would predict a lower target thrombolysis use rate. So instead of providing a single target thrombolysis rate the model provides what might be

considered an 'upper quartile performance' performance given an individual hospital's patient population (which includes both time of arrival and clinical aspects).

Specific points in text:

'The pathway simulation model is coded in Python/NumPy and is based on sampling from distributions based on real-world data':

The second aspect of the sentence seems vague, What data they are referring to? if SSNAP then I think it needs to be re worded, if not there is no reference.

That has been re-worded to make a clearer link to the 'data' section above.

'For machine learning, only those patients with a completed NIH Stroke Scale and who had at least 30 minutes left to give thrombolysis were used (1,862)':

As far as I can tell even in the appendices there is no actual breakdown of the data. So no tabulated ranges for the machine learning model such as age 17-90 etc. So it is hard to say whether this smaller dataset used for this section is actually representative. They trained the model with this data based on the exclusion criteria above, but we don't know whether the dataset they are left with is actually a good representative.

Thank you for the suggestion - we have now added a table of patient characteristics for the two groups. This is in the supplement. The data in the smaller group (those with 30 minutes left to treat) is the most relevant group for the machine model, as patients known to be outside of allowable onset-to-treatment time will not be considered for thrombolysis. This does cause some differences between the two groups; for example, fewer patients aged 80+ are in the '30 minutes left to treat group' because they have a shorter allowable onset-to-treatment time (180 minutes compared to 270 minutes for patients aged under 80).

'the machine learning model allows for differences in patient populations between hospitals (for example reflecting different age demographics between different regions), and allows for differences in real life (rather than clinical trial) decision making':

Without having any idea of the original data variation it is hard to justify this point.

Thank you for the suggestion - we have now added a table to the supplement showing key patient characteristics (for those with time left to treat) for each hospital. An example of differences in populations is that the proportion of patients aged 80+ ranges from 25% to 42%, and average NIHSS on arrival ranges from 9.0 to 11.8. This aligns with the matrix of predicted thrombolysis results in the main paper where the hospitals with the most patients aged 80+ (hospitals 2 & 7) have the lowest predicted thrombolysis rate no matter which hospital the machine learning model was trained on., showing that the machine learning model is taking into account differences in population.

General comment: Thank you for your detailed questions. This has led to significantly more detail being added to the supplement, which we hope will be useful to you and all potential readers.

Reviewer: 2

Reviewer Name: Haris Kamal

Institution and Country: University of Texas at Houston, USA

Please state any competing interests or state 'None declared': None declared

Please leave your comments for the authors below

The authors present a method to qualitatively analyze the application of clinical pathway simulation in machine learning using real-time data to identify important factors to improve use and timing of thrombolysis in acute stroke care hospitals.

The models tried appear to be a good start but perhaps the authors could further validate these models with external data sources- a point which should be mentioned in the discussions section.

Thank you – we have added a point in the strengths and weaknesses that this study is limited in its scope, and that a larger study is recommended (and which we hope to be performing).

Reviewer: 3

Reviewer Name: Andre Kemmling

Institution and Country: Department of Neuroradiology, University Hospital Munster, Germany Please state any competing interests or state 'None declared': None

Please leave your comments for the authors below

Authors use pathway simulation in acute stroke triage with regard to completion of iv thrombolysis. 7864 records with 12 parameters were analyzed. Authors also use machine learning for iv thrombolysis prediction (1862 records with 50 parameters).

Included parameter primarily describe onset time intervals, severity of symptoms and comorbidities. Results show, how onset time intervals affect thrombolysis use in prediction model. across 7 Hospitals.

Results and conclusion may be regarded as proof of concept. Major variables are missing to actually relate to stroke triage reality and the list of parameters is far too simplistic to draw conclusions. Any variable that encodes potential iv rTPA contraindication would affect the model. Imaging parameters such as extent of the early infarct (e.g. by ASPECTS) may significantly affect thrombolysis rates. Also presence of large vessel occlusion may be relevant.

Thank for your comments. We would agree that this study may be considered a proof of concept study. We hope to continue to a larger study, which would be necessary before adoption. We also agree that there are features absent from our model, and this largely explains the accuracy level of the machine modelling of 82%, (in cross-validation, evaluating a test set of data that was not used to train the model), rather than having 90%+ accuracy which we would expect to achieve with more features.

We have enhanced the machine learning validation section in the supplement. That shows that with this set of features we attain 82% accuracy in prediction in use of thrombolysis (when there is time left to treat), which we believe to be high enough to start to draw useful conclusions.

However, we have based our model on SSNAP data because that is relevant to the purpose of the model, which is to aid the national clinical audit.

We have extended the weaknesses in the strengths and weakness section to reflect that, though we have a lot of clinical features available in the SSNAP data set, they do not represent everything about each individual patient. As the SSNAP data set grows (e.g. with more detail on advanced imaging) we would expect the accuracy of the model to continue to increase, though we believe it is already at a useful level for audit review, where we are predicting overall thrombolysis use given a certain population of patients.

We have also mentioned the focus solely on IVT in the weaknesses section, with a note that the principles of the model should also be applicable to use of thrombectomy.

On 'Any variable that encodes potential iv rTPA contraindication', we exclude from the model the SSNAP fields on the reason that thrombolysis was not given (as that would indeed encode the answer directly in to the data, and would give a spurious 100% accuracy). We use only data that would be present before the decision to thrombolysate was made. We do however want to include *potential* contraindications to thrombolysis (as they help reveal differences in decision-making between different hospital), but not any data that has a 100% correlation with the decision to administer thrombolysis.

Authors should address these limitations extensively and explain how these may be overcome and then improve their method.

Authors should use more appropriate up to date references, e.g. 22 Aoki J, Kimura K, Iguchi Y, et al. FLAIR can estimate the onset time in acute ischemic stroke patients. J Neurol Sci 2010;293:39–44. doi:10.1016/j.jns.2010.03.011 references by Thomalla et al are more extensive and recent, in particular with respect to recent NEJM trials with use of MRI to initiate thrombolysis with unknown time of onset.

Thank you – we have included that paper, and the point of selecting patients by advanced imaging, in the discussion and in the strengths and weaknesses section, highlighting that our model currently focusses on those within the 'normal' time limits for thrombolysis. But again, as the SSNAP data set

grows to include these refinements to the selection of patients, and as these refinements become common practice, we would plan to extend the scope of the model.

Reviewer: 4

Reviewer Name: Vikram Puri

Institution and Country: Researcher, DuyTan University, Vietnam

Please state any competing interests or state 'None declared': No competing Interest

Please leave your comments for the authors below

1 . Overall Paper is well written but it require two following revisions :

a. need more references and paper because authors merged the Introduction and related studies on one section

Sorry, we are not too sure what is required here, but I hope the significant enhancement of the supplement, including references to methodology help.

b. They have applied the machine learning and pathway learning so they have to append the results and also some mathematical parameters to increase the visibility of results

Thank you. We have added significantly to the supplement to give more results, and have increased the validation in the main paper itself. We have also added more methodology, both to the paper but mostly in the supplement.

Reviewer: 5

Reviewer Name: Ana Timóteo

Institution and Country: Santa Marta Hospital, Lisbon, Portugal

Please state any competing interests or state 'None declared': None declared

Please leave your comments for the authors below

I have been asked to review specifically the statistics section.

In this paper, there is no specific statistical section. Statistics are very briefly explained in the data and results section. The methodology used by the authors is not usual and it requires a more detailed explanation for the common reader to understand it. It is not clear at all how do the authors get to the figures presented in the paper.

Thank you. We have substantially increased the detail of the methodology (and have added more detailed background data) in the supplement. We have also increased detail on validation of the model (especially in the supplement).

A recommend a detailed statistics section to be included.

Because there are two methods being combined we have tried to place the relevant statistics with each model, but I hope there is enough detail now included (mostly in the supplement) to satisfy.

Reviewer: 6

Reviewer Name: George Vassilacopoulos

Institution and Country: UNIVERSITY OF PIRAEUS, GREECE

Please state any competing interests or state 'None declared': None declared.

Please leave your comments for the authors below

This paper reports on the potential of using simulation and machine learning to enhance the output of the clinical audits recorded by the Sentinel Stroke National Audit Programme (SSNAP) of the NHS England. In particular, the paper focuses on the acute stroke pathway and clinical decision making leading to the use of thrombolysis for the treatment of acute stroke, the only licensed drug treatment for acute stroke, and one that is critically time-dependent with little or no benefit after 4.5 hours from stroke onset.

Essentially, the paper extends previous work on stroke thrombolysis by using both pathway simulation and machine learning modelling to include factors other than door-to-needle times, with special focus on differences in clinical decision making as analysed and modelled with machine learning techniques, so that to develop an open source modelling framework that is fast enough to run routine analysis at national level. The paper is very well presented, with regard to the sound methods (both

simulation and machine learning modelling) used and to the thorough discussion cited, for those interested in the particular clinical field. In addition, it is a significant contribution to the literature on the wider field of healthcare analytics using national registry and/or EMR data. Therefore, I would strongly recommend that the paper is accepted for publication in its present form. Thank you! We hope the revision is even better.

VERSION 2 – REVIEW

REVIEWER	Ana Timóteo Santa Marta Hospital, Lisbon, Portugal
REVIEW RETURNED	07-Aug-2019
GENERAL COMMENTS	The manuscript has been substantially improved in this revision.